# Advancing Multimodal Large Language Models with Quantization-Aware Scale Learning for Efficient Adaptation

## ABSTRACT

This paper presents the first study to explore the potential of parameter quantization for multimodal large language models to alleviate the significant resource constraint encountered during vision-language instruction tuning. We introduce a Quantization-aware Scale LeArning method based on multimodal Warmup, termed QS-LAW. This method is grounded in two key innovations: (1) The learning of group-wise scale factors for quantized LLM weights to mitigate the quantization error arising from activation outliers and achieve more effective vision-language instruction tuning; (2) The implementation of a multimodal warmup that progressively integrates linguistic and multimodal training samples, thereby preventing overfitting of the quantized model to multimodal data while ensuring stable adaptation of multimodal large language models to downstream vision-language tasks. Extensive experiments demonstrate that models quantized by QSLAW perform on par with, or even surpass, their full-precision counterparts, while facilitating up to 1.4 times reduction in VL tuning time and GPU consumption.

## CCS CONCEPTS

• **Computing methodologies → Neural networks**.

## KEYWORDS

Multimodal Large Language Models, Efficient Adaptation, Effective Quantization

## 1 INTRODUCTION

The remarkable performance of large language models (LLMs) has been well-established in recent literature [4, 9, 36, 37, 40], sparking a growing interest in the development of multimodal large language models (MLLMs) [2, 3, 5, 24, 28, 32, 43]. This burgeoning field has led to substantial progress in a wide array of vision-language (VL) tasks. To accomplish this, contemporary MLLMs primarily utilize multimodal instruction following examples for VL instruction tuning and adopt modular architectures [2, 21, 24, 28] to transform visual features into the word embedding space of the LLM. This innovative approach enables LLMs to execute multimodal tasks in an autoregressive fashion. One notable example of this technique is LLaVA [24], which employs a linear projection layer to bridge the gap between the visual encoder and the LLM. By doing so, LLaVA fully harnesses the power of pre-trained LLMs, thereby significantly

**Unpublished working draft. Not for distribution.**

Table 1: Cost and accuracy over various VL instruction tuning paradigms on ScienceQA. The symbol "†" denotes advanced memory-saving strategies, while "OOM" indicates GPU memory exhaustion. Results are evaluated using 4 A800 GPUs.

| Methods | #T-Params | Memory (GB) | Time (hours) | Average (%) |
|---|---|---|---|---|
| LLaVA [28] | 13B | OOM | N/A | N/A |
| LLaVA† [28] | 13B | 71.54 | 8.75 | 90.92 |
| QLoRA | 500.70M | 66.92 | 6.12 | 86.96 |
| QSLAW | 84.25M | 66.52 | 5.76 | **91.04** |

enhancing its visual comprehension capabilities. This seamless integration of visual and linguistic information highlights the potential for MLLMs to revolutionize the field of artificial intelligence and drives further advancements in multimodal tasks.

Despite the advancements, the current VL instruction tuning for MLLMs exhibits considerable redundancy in terms of computation and memory burden. This limitation primarily stems from the inherently large size of LLMs compared to other components within MLLM architectures. For instance, LLaVA-13B fully fine-tunes the entire LLM during VL instruction tuning, often requiring hundreds of GPU hours [28]. Although recent efforts have introduced more efficient adapters and the freezing of LLMs to reduce training overheads [15, 32], VL tuning within current MLLM frameworks still demands substantial memory usage and computational resources, necessitating at least 8 NVIDIA Tesla A100 GPUs [32]. This poses great challenges to the rapid adaptation of LLMs for cross-modal tasks, particularly in situations characterized by limited training resources and needs for on-the-fly, task-specific tuning.

To address this constraint, this paper explores the potential of parameter quantization for MLLMs, aiming to alleviate the extensive training demands encountered during VL instruction tuning while preserving the original performance. Quantization, a network compression technique, transforms the full-precision weights into low-bit representations, consequently reducing both computational load and storage requirement. It has been adopted for parameter-efficient fine-tuning (PEFT) of LLMs [12, 17, 44], notably in QLoRA [12], which quantizes each linear layer's weights into a 4-bit NormalFormat (NF) datatype and uses the low-rank adapter (LoRA) [20] for fine-tuning. Owing to the lightweight quantized LLM and a minimal set of trainable parameters within the LoRA module, QLoRA can facilitate LLaMA-65B fine-tuning on a single 48GB GPU without sacrificing chat performance [12].

A potential strategy to consider is implementing the previously discussed PEFT method to facilitate VL instruction tuning for MLLMs. We conduct an experiment in Table 1 to analyze its efficacy. As can be seen, utilizing QLoRA to quantize LLM weights to 4-bit can significantly reduce both GPU memory consumption and time overhead. Regrettably, QLoRA inflicts a considerable performance impairment on multimodal tasks, with almost a 4% accuracy decrease on ScienceQA [31], despite its capacity to attain parity with full-precision performance in language tasks [19, 41]. This incongruity prompts

us to investigate the effects of quantization on MLLMs during VL instruction tuning in greater depth. Consequently, we examine the activation distribution within intermediate layers of LLMs, focusing on language and multimodal image data, as depicted in Figure 1. A noticeable percentage of activations emerge as outliers, displaying significant deviations in magnitude, which poses a substantial challenge for MLLMs quantization. Minor quantization errors may accumulate and interact with these activation outliers, ultimately resulting in irreversibly distorted outputs [11, 27]. Furthermore, the density and frequency of activation outlier markedly increase in multimodal inputs compared to unimodal language inputs. This observation elucidates the performance deterioration of QLoRA in VL instruction tuning for MLLMs, as its adopted NF4 datatype only pursues equating the quantity of values across all quantization bins from the weight tensor and causes severe information loss, making it hard for LoRA to accommodate activation outliers.

To address this limitation, we employ quantization-aware scale learning instead of using LoRA to fine-tune the quantized ML-LLM. Specifically, we divide the weights into multiple quantization groups, each assigned a learnable scale factor. This scale learning approach effectively reduces quantization errors within each group, particularly in cases where activations exhibit outlier characteristics at certain positions. Furthermore, we adopt uniform quantization instead of forcing an equal number of weights in each quantization bin like NF4 and initialize the quantized weights with Omni-Quant [38], an LLM uniform quantization method that employs weight clipping to mitigate the quantization difficulty occurring in language tasks, rather than relying solely on the probability density of the weights as in NF4. Compared to LoRA, our proposed scale learning method offers several distinct advantages. First and foremost, LoRA fine-tuning targets the output of the entire layer, which fails to adaptively minimize the quantization errors at outlier positions. Additionally, scale learning exhibits significantly higher efficiency than LoRA. For example, with a group size of 128, the introduced parameters for scale learning amount to only 16.83% of those required for fine-tuning with LoRA, thereby enabling more efficient VL instruction tuning for MLLMs.

Next, we explore the data construction for scale learning. Relying solely on multimodal data for training scaling tends to cause the LLM to overfit to downstream tasks, subsequently diminishing its inherent linguistic proficiency. On the other hand, using a heterogeneous mix of language and multimodal data compromises the efficiency of VL tuning, as MLLMs fail to receive adequate multimodal instructional guidance during the initial stages of training. To address this issue, we introduce a novel modality-aware warmup method, which utilizes only multimodal data during the early phase of VL tuning and subsequently incorporates language data for scale learning. This ensures that MLLMs receive precise multimodal instructional supervision and avoids potential overfitting of the LLM backbone on multimodal data in the later stages of training, thereby preserving its original linguistic knowledge.

Our Quantization-aware Scale LeArning based on multimodal Warmup, termed QSLAW, is demonstrated to be effective for efficient MLLM instruction tuning across various VL tasks. For instance, QSLAW achieves 91.04% accuracy on ScienceQA with LLaVA-13B, representing a 4.08% gain compared to the 86.96% achieved by

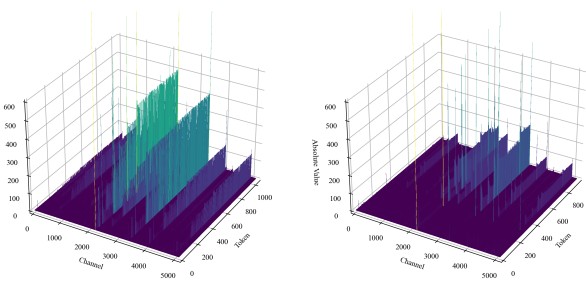

**Figure 1: Absolute magnitude of the input activation in one LLaVA-13B block. Left (image and text tokens) exhibits a larger scale in activation compared to the right (only text tokens).**

QLoRA, which even outperforms the full-precision fine-tuned LLaVA-13B (91.04 for QSLAW *v.s.* 90.92 for full-precision). Our contributions include:

- We undertake the pioneering exploration of MLLMs quantization and utilize scale learning to alleviate the quantization challenges arising from the frequent occurrence of activation outliers inherent to MLLM quantization.
- We introduce a modality-aware warmup called multimodal warmup to prevent the quantized model from overfitting to multimodal data while ensuring stable adaptation of MLLMs to downstreams.
- Extensive experiments validate that QSLAW significantly reduces training time and memory footprint for VL instruction tuning while maintaining state-of-the-art performance.

## 2 RELATED WORK

### 2.1 Model Quantization

Quantization methods can be broadly classified into quantization-aware training (QAT) and post-training quantization (PTQ). QAT relies on the complete training data to fine-tune the quantized model in line with the pre-training phases [8, 14, 29, 30, 39]. PACT [8] employs trainable activation clipping parameters to determine the appropriate quantization step size, while LSQ [14] directly optimizes the step size through carefully designed gradient scaling. N2UQ [29] incorporates a set of learnable thresholds to achieve nonuniform-to-uniform quantization and utilizes a generalized straight-through estimator for optimization. Although QAT exhibits promising performance retention, it suffers the need for training weights and quantization parameters on the full dataset. In contrast, PTQ can efficiently perform quantization with significantly less data and resources. Adaround [35] introduces a learnable variable for each weight and optimizes them layer by layer using a soft relaxation. BRECQ [25] extends the PTQ framework to block-wise optimization using second-order information. Qdrop [42] further incorporates activation quantization and suggests randomly dropping quantized activation to enhance the flatness of quantized models.

Although these methods are highly efficient for CNNs, they cannot be extended to LLMs due to the difficulty of optimizing the vast parameter space with limited samples. GPTQ [16] is the first attempt to implement PTQ on models with billions of parameters, utilizing second-order information to compensate for quantization

error. LLM.int8() [11] highlights significant errors in quantization caused by activation outliers within LLMs, leading to the proposal of a mixed-precision quantization approach. AWQ [27] proposes input channel scaling based on activation to protect essential weights. OmniQuant [38] designs learnable weight clipping and learnable equivalent transformation, making them differentiable to quantize LLMs with gradient optimization. AffineQuant [33] further proposes directly optimizing equivalent affine transformations. These methods aim to reduce the memory footprint of LLMs during inference. QLoRA [17] is the first to explore reducing memory footprint during training through quantization and LoRA [20]. LoftQ [26] and LQ-LoRA [17] alternate between quantization and singular value decomposition to find a suitable initialization for LoRA. QA-LoRA [44] proposes balancing degrees of freedom between quantization and adaptation with group-wise down projection of LoRA. These methods aim to enhance LoRA based on the NF data type, contributing orthogonally to our approach. Our method falls within the realm of quantization for LLMs. Unlike the aforementioned methods, we specifically address the quantization challenges in MLLMs, where additional visual inputs can influence the activation distribution and pose a new challenge. Also, we are the first to explore quantization challenges for MLLMs.

## 2.2 Multimodal Large Language Models

Traditional VL instruction tuning commonly employs various task-related losses, including image-text contrastive loss, image-text matching loss, and language modeling loss, to supervise the training of both visual and language branches. To compute these losses, it is typically necessary to perform multiple forward passes on the image-text pairs, consuming thousands of GPU hours. However, with the emergence of large language models (LLMs), the paradigm of VL tuning has shifted towards treating LLMs as a universal interface and adopting a modular structure to align representations from vision with LLMs. In these approaches, LLMs and the modular structure are trained on multimodal examples using a simple cross-entropy loss. Recent advances in this area include Flamingo [2], which introduces the Perceiver Resampler as the modular structure, and BLIP2 [24], which proposes a lightweight Q-Former to align different modalities. LLaVA [28] employs a simple MLP as a modular structure and introduces VL instruction tuning, enabling LLMs to execute multimodal tasks in an autoregressive manner. Despite these advancements, the current VL instruction tuning for massive MLLMs remains expensive. For example, LLaVA-13B fully fine-tunes the entire LLM during VL instruction tuning, often requiring hundreds of GPU hours. LaVIN [32], which utilizes an adapter to achieve parameter-efficient VL instruction tuning, still necessitates at least eight NVIDIA Tesla A100 GPUs. Our method is designed to alleviate the extensive training demands in VL instruction tuning while preserving the original performance. And we are the first to achieve this for VL instruction tuning by employing quantization with a minimal set of trainable parameters.

## 3 METHODOLOGY

### 3.1 Preliminary

The objective of vision-language (VL) instruction tuning for MLLMs is to adapt an LLM backbone from processing unimodal text data

to encompassing multimodal data. Specifically, given a multimodal instruction following example that consists of an image $\mathbf{I} \in \mathbb{R}^{h \times w \times 3}$ and a text sequence $\mathbf{T} \in \mathbb{R}^l$, the image $\mathbf{I}$ is initially fed into an image encoder, typically a pre-trained vision transformer [13], to extract the informative visual representation as:

$$\mathbf{F_I} = f_{\theta_I}(\mathbf{I}), \tag{1}$$

where $\theta_I$ represents the encoder's parameters. Then, the visual representation is projected to the word embedding space of LLMs through a modular structure parameterized by $\theta_a$:

$$\mathbf{F_I}' = f_{\theta_a}(\mathbf{F_I}). \tag{2}$$

Subsequently, the LLM with pre-trained weights $\mathbf{W}$ receives the embedded image feature and the text sequence $\mathbf{T}$ to generate a probability distribution $\mathbf{P} \in \mathbb{R}^{L \times N}$ for each word in the target response $\mathbf{R} \in \mathbb{R}^L$:

$$\mathbf{P} = g_{\mathbf{W}}(\mathbf{F_I}', \mathbf{E_T}), \tag{3}$$

where $\mathbf{E_T} = f_{\theta_T}(\mathbf{T})$ is the word embedding of the input text sequence and $N$ denotes the vocabulary size of the pretrained LLM.

Finally, the modular structure and LLM are jointly fine-tuned by minimizing the cross-entropy loss, which can be formulated as:

$$\mathcal{L} = -\sum_{i=1}^{L} \log \mathbf{P}_{i,j}, \tag{4}$$

where $j$ represents the position of $\mathbf{R}_i$ in the vocabulary.

Albeit the efficacy, it requires considerable computational resources and memory usage, mainly streaming from the significantly large parameters in LLMs. Although recent advancements [12, 23, 32, 48] have shown the potential of freezing the LLM backbone to eliminate partial backward costs of the LLM backbone, existing VL tuning frameworks still require a minimum of 8 NVIDIA Tesla A100 GPUs. This necessity poses significant challenges to the efficient adaptation of LLMs to cross-modal tasks, particularly in situations characterized by limited training resources and the need for on-the-fly, task-specific fine-tuning.

### 3.2 The Potential of LLMs Quantization

Quantizing the parameters of an LLM backbone into lower-bit representations offers a promising solution to the above problem and has shown remarkable efficacy in traditional unimodal PEFT scenarios for LLMs [12, 17, 22, 26]. Therefore, we initiate an exploratory investigation into the potential of quantization for MLLMs, starting with a trial using the QLoRA [12] in PEFT contexts.

Specifically, QLoRA compresses the normalized weight $\hat{\mathbf{W}}$ into quantized weight $\overline{\mathbf{W}}$ with 4-bit NormalFloat (NF) format $q_i$ as:

$$\overline{\mathbf{W}} = \arg\min_{q_i}|\hat{\mathbf{W}} - q_i|, \tag{5}$$

where $\hat{\mathbf{W}} = \frac{\mathbf{W}}{max(abs(\mathbf{W}))}$ and $q_i$ in 4-bit NF format is:

$$q_i = \frac{1}{2}\big( Q(\frac{i}{17}) + Q(\frac{i+1}{17}) \big), \tag{6}$$

where $Q(\cdot)$ is the quantile function of the standard Gaussian distribution $\mathcal{N}(0, 1)$. After the NF4 initialization, the quantized LLM is frozen, and the low-rank adapter (LoRA) [20] is then utilized for transfer learning on downstream tasks. It has been widely demonstrated in the literature [6, 18, 46, 47] that QLoRA can maintain the

performance of LLMs on downstream language tasks with 4-bit quantization, significantly reducing the training burden and memory costs. Regrettably, when we employ QLoRA to enhance the efficiency of MLLM training, a nearly 4% decrease in accuracy is observed on ScienceQA, as illustrated in Table 1. Upon analysis, the NF format described in Eq. (6) only ensures an equal quantity of weight values across each quantization bin but overlooks activation outliers, which are common in LLMs. Furthermore, our observation in Figure 1 indicates that the density and frequency of activation outliers significantly increase with multimodal inputs compared to unimodal language inputs. Consequently, minor quantization errors may accumulate and interact with these activation outliers, ultimately leading to irreversible output distortion. In summary, although quantization holds considerable potential for alleviating the massive burden of MLLM tuning, the need to address the dense activation outlier phenomenon in multimodal scenarios remains pressing.

### 3.3 QSLAW

We formally present our Quantizration-aware Scale LeArning based on multimodal Warmup (QSLAW) method, specifically designed for efficient Visual Language (VL) instruction tuning in MLLMs. QSLAW addresses the challenges associated with MLLMs quantization from two aspects: (1) it uniquely learns scale factors for different weight groups, reducing the quantization error resulting from activation outliers and demonstrates to be more effective for VL instruction tuning on quantized LLMs; (2) it employs a modality-aware warmup strategy called multimodal warmup, which blends linguistic and multimodal training samples, thus preventing the quantized model from overfitting to multimodal data while ensuring a stable adaptation to the target VL tasks.

**Quantization-Aware Scale Learning.** During the VL instruction tuning process, we assign learnable group-wise scale factors **s** to the LLM weights **W** as follows:

$$\hat{\mathbf{W}} = \frac{\mathbf{W}}{\mathbf{s}} \tag{7}$$

And then uniform quantization is utilized to convert $\hat{\mathbf{W}}$ into pseudo-quantized weight $\tilde{\mathbf{W}}$:

$$\tilde{\mathbf{W}} = \Delta \times (\text{clamp}(\lfloor \frac{\hat{\mathbf{W}}}{\Delta} \rceil + zp, 0, 2^k - 1)) - zp, \tag{8}$$

where $\lfloor \cdot \rceil$ denotes the round-to-nearest integer operation and $k$ represents the quantization bit. $\Delta$ and $zp$ are the quantization step-size and zero-point, respectively.

In this approach, **W** is divided into multiple groups, with each group of weights being scaled by a single factor. By learning the scale factor under the guidance of Eq. (4), the quantization error within each weight group can be effectively minimized towards downstream tasks, particularly for groups containing activations outlier. To clarify, an appropriate scaling factor allows for the rescaling of weights into a quantile range that reduces the output perturbation when interacting with activation outlier exhibiting significant deviation magnitudes and is more suitable for quantized LLMs to transfer into VL tasks with VL instruction tuning. In contrast, LoRA is unable to effectively mitigate such quantization errors caused by activation outliers, as it conducts fine-tuning in a coarse-grained, global manner. Importantly, as demonstrated in Table 1,

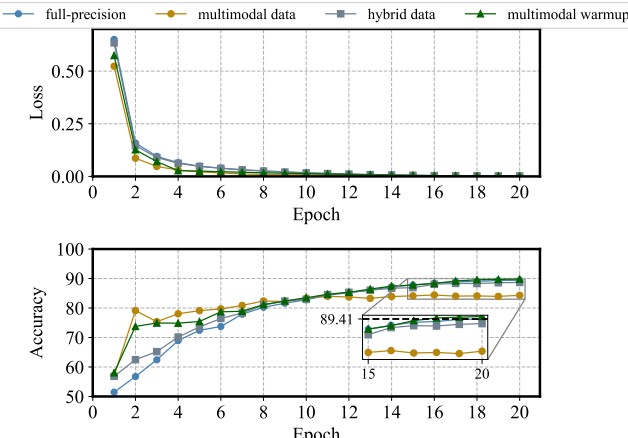

Figure 2: Loss and accuracy curves of training scaling with different strategies on ScienceQA. Solely utilizing multimodal data for training scaling tends to lead the LLM overfitted to downstream tasks. This is evidenced by a rapid decrease in loss but the accuracy remain mediocre.

the parameter count of the scale learning is substantially lower than that of the LoRA module, making our scale learning suitable for an efficient VL instruction tuning.

**Modality-aware Warmup.** As discussed in Sec. 3.1, MLLMs fine-tune their parameters using instruction samples that encompass both images and textual content. Unfortunately, we find that training the scale factor for quantization using the same dataset can result in an overfitting issue for the LLM backbone. As illustrated in Figure 2, scale learning on purely multimodal data leads to a rapid decrease in loss while the final model accuracy, paradoxically, fails to outperform the full-precision counterpart. This overfitting phenomenon is understandable, given that the LLM's pre-training was solely based on linguistic data. Consequently, conducting quantization-aware scale training exclusively on multimodal data can impair the inherent linguistic capabilities of the LLM, which are of paramount importance to serve as a language backbone for multimodal adaptation.

An intuitive solution to this overfitting problem involves the integration of linguistic data to jointly guide the scale learning process. Consequently, we supplement the existing multimodal data with the WikiText dataset [34], thereby creating a hybrid dataset specifically designed for scale learning. Figure 2 illustrates the trajectories of loss and accuracy. While effective mitigation of overfitting is observed, the adaptive performance of the multimodal approach still falls short when compared to its full-precision counterpar. We attribute this outcome to an early-stage underfitting of the LLMs with respect to the multimodal data. Specifically, the MLLMs parameters, such as the scale factor and modular structure, are randomly initialized at the start of training, and the interference of linguistic supervision at this stage hinders the model's ability to fit the multimodal data, resulting in suboptimal performance.

To address the complex interplay between overfitting and underfitting, we equip QSLAW with a multimodal warmup data sampling strategy. More specifically, during the initial $\eta$ iterations of VL instruction fine-tuning, we exclusively utilize multimodal data

**Table 2: Quantitative accuracy on ScienceQA test dataset. Question classes: NAT = natural science, SOC = social science, LAN = language science, TXT = text context, IMG = image context, NO = no context, G1-6 = grades 1-6, G7-12 = grades 7-12. The symbol "†" denotes a larger rank used for LoRA and the best results in each class are underlined.**

| Method | Subject | | | Context Modality | | | Grade | | Average |
|---|---|---|---|---|---|---|---|---|---|
| | NAT | SOC | LAN | TXT | IMG | NO | G1-6 | G7-12 | |
| *Zero-shot & few-shot* representative methods with performance reported in the literature | | | | | | | | | |
| Human [31] | 90.23 | 84.97 | 87.48 | 89.60 | 87.50 | 88.10 | 91.59 | 82.42 | 88.40 |
| GPT-3.5 [31] | 74.64 | 69.74 | 76.00 | 74.44 | 67.28 | 77.42 | 76.80 | 68.89 | 73.97 |
| GPT-3.5 w/ CoT [48] | 75.44 | 70.87 | 78.09 | 74.68 | 67.43 | 79.93 | 78.23 | 69.68 | 75.17 |
| *Two-stage* representative methods with performance reported in the literature | | | | | | | | | |
| LLaVA-13B [28] | 90.36 | 95.95 | 88.00 | 89.49 | 88.00 | 90.66 | 90.93 | 90.90 | 90.92 |
| LLaVA-13B-QLoRA | 74.20 | 79.19 | 69.55 | 74.10 | 70.40 | 72.47 | 75.51 | 71.39 | 74.04 |
| LLaVA-13B-QLoRA† | 85.48 | 93.59 | 84.64 | 84.56 | 83.94 | 86.90 | 87.96 | 85.17 | 86.96 |
| LLaVA-13B-QSLAW (Ours) | 83.26 | 91.79 | 80.00 | 82.80 | 81.30 | 83.07 | 86.01 | 80.95 | **84.20** |
| LLaVA-13B-QSLAW†(Ours) | 91.30 | 96.06 | 86.45 | 90.22 | 89.34 | 89.34 | 91.63 | 89.98 | **91.04** |
| *One-stage* representative methods with performance reported in the literature | | | | | | | | | |
| LLaMA-Adapter [48] | 84.37 | 88.30 | 84.36 | 83.72 | 80.32 | 86.90 | 85.83 | 84.05 | 85.19 |
| LaVIN-7B [32] | 89.25 | 94.94 | 85.24 | 88.51 | 87.46 | 88.08 | 90.16 | 88.07 | 89.41 |
| LaVIN-7B-QLoRA | 87.61 | 94.04 | 85.18 | 86.51 | 85.62 | 88.43 | 89.02 | 83.08 | 88.27 |
| LaVIN-7B-QSLAW (Ours) | 90.23 | 93.59 | 85.82 | 89.54 | 87.75 | 88.71 | 90.75 | 88.07 | **89.79** |

pairs for scale learning. Subsequently, we incorporate linguistic text sequences extracted from the WikiText dataset to facilitate hybrid-data training. This warmup approach ensures accurate multimodal instructional supervision for the MLLMs during the initial training iterations, while simultaneously circumventing potential overfitting of the LLM backbone on multimodal data to preserve its inherent linguistic capabilities. Consequently, as demonstrated in Figure 2, the proposed multimodal warmup method effectively rivals the accuracy of quantization-aware scale learning compared with its full-precision counterpart.

In addition to the two components mentioned above, our proposed QSLAW method initializes $\Delta$ and $zp$ in Eq. (5) using OmniQuant [38], a post-training quantization method for LLMs that employs weight clipping to mitigate the quantization challenge in language tasks, rather than solely relying on the probability density of the weights as in NF4. And it is worth mentioning that our method is not constrained by the quantization method or initialization and can achieve consistent performance improvements compared with Lora as discussed in Sec. 4.3.2. Moreover, it is of crucial importance to realize that QSLAW is orthogonal to most of off-the-shelf MLLMs paradigms and can be seamlessly integrated to enhance their efficiency during the VL instruction tuning, which will be quantitatively demonstrated in the following experimental section.

## 4 EXPERIMENTATION

### 4.1 Experimental Settings

*4.1.1 Networks and Datasets.* To validate the effectiveness of our approach, we select two types of MLLMs: LLaVA [28], which employs a two-stage full fine-tuning strategy, and LaVIN [32], which utilizes one-stage parameter-efficient fine-tuning strategy. We evaluate performance in line with most multimodal LLMs [10, 28, 32, 45],

focusing on visual reasoning and instruction-following capabilities. For a straightforward comparison, we follow the precedent set by LLaVA and LaVIN to choose the ScienceQA dataset [31] for visual reasoning. The dataset, split into *train*, *val*, and *test*, spans diverse domains, including natural science, language science, and social science, and consists of both text-image and text-only inputs. We report the average accuracy on its *test* split. For instruction-following, we construct a multimodal ChatBot using LLaVA trained with LLaVA-80k [28]. LLaVA-80k is a high-quality vision-language instruction-following dataset generated by ChatGPT/GPT-4 [1]. The responses from the ChatBot will be evaluated by GPT-4, with higher-quality responses receiving a score ranging from 1 to 10.

*4.1.2 Implementation Details.* Following papers [28, 32], we adopt the ViT-L/14 in CLIP as the image encoder. For LaVIN and LLaVA, we use two MLP layers with a hidden dimension of 128 and a simple linear layer as modular structure, respectively. For LLMs, we employ LLaMA-7B [40] and Vicuna-13B [7]. All parameter settings strictly adhere to the LLaVA and LaVIN papers, except for the 2 training epochs with a batch size of 64, and a 1:1 hybrid training dataset comprising WikiText and downstream data for scale learning.[1]

### 4.2 Main Results

*4.2.1 ScienceQA.* We categorize MLLMs into one-stage parameter-efficient and two-stage full fine-tuning and select a renowned model for each category to validate our method's performance. Quantitative results on ScienceQA are presented in Table 2. Our approach significantly enhances quantization transfer performance on multimodal tasks, showing consistent improvement across all question classes compared to QLoRA. For LLaVA, our QSLAW achieves 84.20% accuracy, a 10.16% gain over QLoRA's 74.04%. With improved settings, our method even outperforms full-precision LLaVA-13B,

---

[1]Code in the supplementary material will be open-source upon paper acceptance.

**Image Caption**

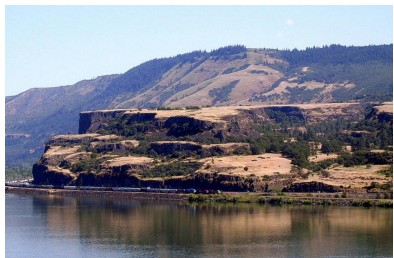

Q: Can you elaborate on the elements of the picture provided?

➤ **QLoRA:** The image shows a beautiful landscape with a lake, mountains, and a valley. The lake is surrounded by mountains, and there is a road running along the side of the lake. The sky is blue and clear, adding to the serene and picturesque atmosphere of the scene. The presence of a car on the road suggests that the area is accessible and possibly a popular destination for tourists or locals to enjoy the scenic views and outdoor activities. The combination of the lake, mountains, and valley creates a breathtaking natural landscape that captures the essence of the region's beauty.

➤ **LLaVA:** The image features a large body of water, possibly a lake or a river, with a mountain range in the background. The water appears to be calm and serene, creating a peaceful atmosphere. The landscape is picturesque, with the mountain range providing a stunning backdrop to the water. The scene is likely to be a popular spot for tourists and locals alike, who may come to enjoy the natural beauty, take photographs, or engage in recreational activities such as boating, fishing, or hiking. The presence of the mountain range also suggests that the area might be suitable for various outdoor activities, such as rock climbing, mountain biking, or camping.

➤ **QSLAW:** The image shows a beautiful landscape with a large body of water, possibly a lake, situated in a valley. The water is surrounded by mountains, creating a picturesque and serene environment. There are also trees in the scene, adding to the natural beauty of the landscape. The sky in the image is blue, which further enhances the overall aesthetic of the picture. The presence of a train in the background adds an element of human activity and transportation to the scene, contrasting with the natural beauty of the landscape.

**Multimodal Reasoning**

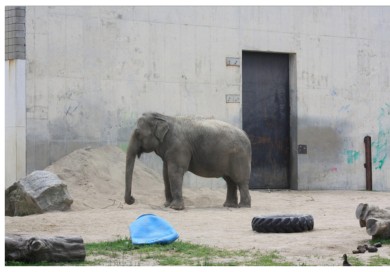

Q: What could be a potential concern for the baby elephant in this setting?

➤ **QLoRA:** A potential concern for the baby elephant in this setting could be its safety and well-being. In the image, the baby elephant is standing in a sandy area with a blue tire and a blue ball nearby. The presence of these objects might pose a risk to the baby elephant, as it could accidentally step on or play with them, potentially causing injuries or harm. Additionally, the baby elephant might be curious and explore the objects, which could lead to further accidents or mishaps. It is essential for the caretakers to ensure the safety and well-being of the baby elephant by providing a secure and appropriate environment for its growth and development.

➤ **LLaVA:** A potential concern for the baby elephant in this setting is the presence of the tire in the enclosure. The tire might pose a risk to the elephant's safety, as it could be a choking hazard or cause injury if the elephant were to play with it or accidentally step on it. Additionally, the tire might not be an appropriate toy for an elephant, as it could be too small or not sturdy enough to withstand the elephant's size and strength. It is essential for the caretakers to ensure that the enclosure is safe and provides appropriate toys and stimulation for the baby elephant's well-being.

➤ **QSLAW:** A potential concern for the baby elephant in this setting is the lack of natural habitat and social interaction with other elephants. In the image, the baby elephant is standing alone in a dirt area with a few toys and a tire nearby. This environment might not provide the baby elephant with the necessary stimulation, socialization, and learning opportunities that it would typically receive in the wild. Additionally, the presence of a tire and other toys suggests that the baby elephant might be in a zoo or a similar enclosure, which could limit its natural behaviors and movement. It is essential for the baby elephant to have access to a more natural habitat and social interactions with other elephants to ensure its proper growth and development.

**Visual Comprehension**

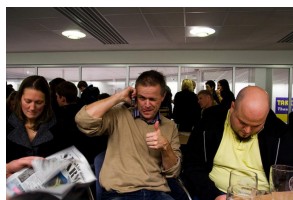

Q: What is the man doing in the image?

➤ **QLoRA:** The man in the image is talking on his cell phone while sitting at a table.

➤ **LLaVA:** The man in the image is talking on his cell phone while sitting at a table.

➤ **QSLAW:** The man in the image is sitting at a table, holding up a cell phone to his ear, and talking on the phone.

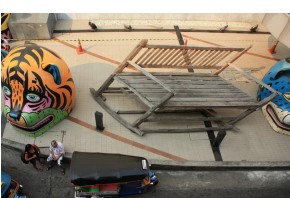

Q: What kind of objects are included in the art installation?

➤ **QLoRA:** The art installation includes a large tiger sculpture and a car.

➤ **LLaVA:** The art installation includes a large orange tiger sculpture and a wooden structure.

➤ **QSLAW:** The art installation includes a large tiger statue, a wooden bench, and a car.

**Figure 3: Comparison among various VL instruction tuning paradigms with the examples under different multimodal instruction-following tasks including visual comprehension, image caption and multimodal reasoning. More detailed parts of the response are marked in red and the misunderstandings in responses are marked in bluesky.**

which fine-tunes the entire LLM on ScienceQA. For LaVIN, our QS-LAW achieves 89.79% accuracy, a 1.46% gain compared to QLoRA's 88.33%, and outperforms full-precision LaVIN-7B.

*4.2.2 ChatBot.* We also present qualitative results to demonstrate the multimodal instruction-following capabilities of models obtained using QSLAW. In Figure 3, we compare various VL instruction tuning paradigms with examples from different multimodal instruction-following tasks, including image captioning, multimodal reasoning and visual comprehension. In more challenging image

captioning tasks requiring both local and global image understanding, QSLAW excels. For a relatively small, distant train in an image, QSLAW correctly identifies it, whereas QLoRA misidentifies it as a car, and LLaVA overlooks it. This misunderstanding leads LLaVA and QLoRA to generate incorrect speculations about the scene, where it is unlikely for tourists to relax or take photos near a railway. Additionally, QSLAW exhibits exceptional multimodal reasoning capabilities. It can infer the absence of social interaction from the environment and the position between the tir and the

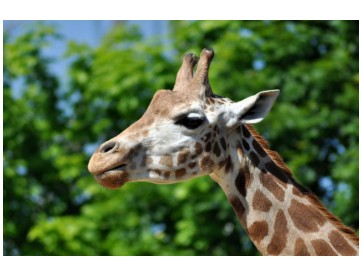

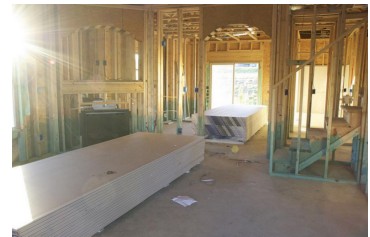

**Q:** What is the main focus of the image?

➤ **QSLAW:** The main focus of the image is a close-up view of a giraffe's face, with its eyes and mouth visible. *[GPT Score: 9]*

➤ **QLoRA:** The main focus of the image is a giraffe standing in a field of trees. *[GPT Score: 6]*

**GPT-4 Evaluation:** QSLAW's response was slightly more detailed, specifying that the close-up view of the giraffe's face includes its eyes and mouth. This additional detail gives QSLAW a higher score, as it provides a more comprehensive description of the image's main focus. QLoRA's response is not as precise, as it focuses on the giraffe standing in a field of trees, which is not the main focus of the image. QSLAW's answer is more helpful and detailed, while QLoRA's answer is less accurate and less relevant to the question asked.

**Q:** What may be a potential obstacle for the house construction in its current state?

➤ **QSLAW:** In the image, the house construction is in its early stages, with the foundation and walls being built. A potential obstacle for the construction in its current state is the lack of proper support and reinforcement for the walls. The walls are made of wood, which may not provide sufficient strength and stability for the structure. *[GPT Score: 7]*

➤ **QLoRA:** A potential obstacle for the house construction in its current state is the presence of a large piece of furniture, such as a couch, in the middle of the room. This furniture might be blocking the path for workers and equipment, making it difficult for them to move around and complete the construction process efficiently. Additionally, the furniture might also be in the way of the electrical outlets, which could cause safety hazards or delays in the installation of electrical systems. *[GPT Score: 3]*

**GPT-4 Evaluation:** QSLAW focused on the lack of proper support and reinforcement for the walls, which is a valid concern but not directly related to the current state of the house construction. QLoRA's response was not relevant to the given context, as it focused on the presence of furniture, which was not mentioned in the context. Therefore, QSLAW receives a score of 7 for providing a helpful and accurate response, while QLoRA receives a score of 3 for providing an irrelevant response.

**Figure 4: GPT-4 scores for QSLAW and QLoRA. Higher score represents high quality and the reasons why QSLAW obtains a higher score are highlighted in red.**

**Table 3: Effect of each component on ScienceQA test dataset. All results are conducted with LaVIN-7B.**

| Settings | NAT | SOC | LAN | TXT | IMG | NO | G1-6 | G7-12 | Average (%) |
|---|---|---|---|---|---|---|---|---|---|
| OmniQuant | 88.06 | 94.15 | 84.36 | 87.49 | 86.61 | 86.90 | 89.57 | 86.22 | $88.38_{(+0.00)}$ |
| + quantization-aware scaling | 84.64 | 89.65 | 80.64 | 83.82 | 81.46 | 83.97 | 86.78 | 80.82 | $84.65_{(-3.73)}$ |
| + hybrid data | 87.26 | 94.83 | 86.64 | 86.56 | 85.77 | 89.41 | 89.98 | 86.35 | $88.68_{(+0.30)}$ |
| + multimodal warmup | 90.23 | 93.59 | 85.82 | 89.54 | 87.75 | 88.71 | 90.75 | 88.07 | $89.79_{(+1.41)}$ |

baby elephant, which is contextually consistent. In contrast, LLaVA and QLoRA merely deduce from the objects in the scene. For simple visual comprehension questions, QSLAW generates detailed and precise responses. For instance, QSLAW provides a more comprehensive description of a man's posture while making a phone call compared to QLoRA and LLaVA. In another image, QSLAW accurately recognizes a wooden structure as a bench and offers a more thorough description of the scene, while QLoRA and LLaVA have omissions. These examples illustrate that our proposed QSLAW in this paper effectively learns visual knowledge and instruction-following abilities during VL instruction tuning.

We also use very strong GPT-4 [1] to evaluate the response quality from QSLAW and our QLoRA. The results are reported in Figure 4. QSLAW performs better than QLoRA, primarily due to its detailed descriptions and superior visual comprehension.[2]

### 4.3 Ablation Studies

*4.3.1 Component Importance.* We examine the effectiveness of each component to provide deeper insights into VL instruction tuning with quantization. Table 3 shows that when LLM is quantized

---
[2]More evaluation results have been illustrated in supplementary material.

by OmniQuant and undergoes VL instruction tuning on ScienceQA like LaVIN, it serves as our baseline and achieves higher accuracy compared to LaVIN-QLoRA due to the consideration of activation outliers. When we introduce quantization-aware scale learning and train it on the same dataset used for VL instruction tuning, the performance drops significantly due to overfitting issues in the LLM backbone. Incorporating linguistic data to guide scale learning alleviates overfitting and improves average accuracy by 0.66% compared to the baseline. Nevertheless, it still lacks effective supervision and exhibits a performance gap compared to full-precision LaVIN (88.54% for hybrid data *v.s.* 89.41% for the full-precision). Our multimodal warmup allows for precise supervision with hybrid data and demonstrates potential beyond models with full-precision VL instruction tuning.

*4.3.2 Quantization Initialization.* We further evaluate QSLAW's performance with different quantization initialization on ScienceQA. In Table 4, we examine the performance of LaVIN-7B under both NF4 and OmniQuant. Our method consistently enhances performance under these two different quantization initializations. Specifically, QSLAW demonstrates an improvement of 0.71% and 1.41% compared to LoRA for NF4 and OmniQuant, respectively. This

**Table 4: Different Quantization initialization method for LoRA and QSLAW. OmniQuant-1 and OmniQuant-2 means the calibration for quantization parameters are conducted on language dataset and hybrid dataset, respectively.**

| Initialization | Average (%) |
|---|---|
| *LoRA is used.* | |
| NF4 (QLoRA) | 88.27 |
| OmniQuant-1 | 88.38 |
| OmniQuant-2 | 88.54 |
| *QSLAW is used.* | |
| NF4 | 88.98 |
| OmniQuant-1 | 89.79 |
| OmniQuant-2 | 89.85 |

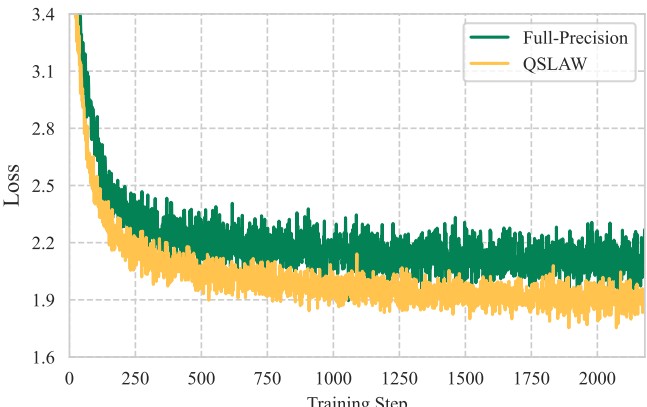

**Figure 5: The training process with different strategies. With our multimodal warmup strategy, the training process exhibits faster and more stable fitting.**

result also validates that, for VL tuning where the density and frequency of activation outlier are markedly increased, the NF4 datatype, which aims to equalize the quantity of values across all quantization bins, is sub-optimal and may negatively impact VL tuning. Moreover, QSLAW outperforms LoRA under both quantization methods, illustrating that our proposed scale learning method is more suitable for VL instruction tuning with quantized LLMs. This can be attributed to QSLAW's ability to effectively adapt to the unique characteristics of each quantization method, ensuring optimal performance in various quantization scenarios.

In conclusion, QSLAW's versatility and adaptability make it a robust and effective solution for VL instruction tuning across different quantization methods, leading to improved performance and more accurate results in multimodal tasks.

*4.3.3 Alignment Effect.* To further elucidate the benefits of the proposed QSLAW method in this paper, we conduct in-depth experiments on a two-stage LLaVA model. This model features a separate stage dedicated to pre-training a modular structure, which allows us to exclude the influence of other trainable parameters. This setup enables us to observe how quantization-aware scale learning

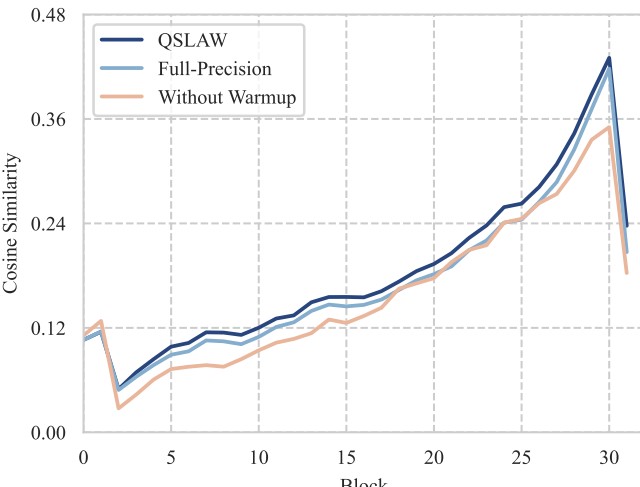

**Figure 6: Block-wise cosine similarity between visual tokens and text tokens under different strategies. QSLAW can help model to align visual and textual tokens.**

enhances the alignment between visual and language modalities, ultimately leading to improved performance results.

As depicted in Figure 5, QSLAW can stabilize and accelerate the training process for the modular structure. We also calculate the pair-wise consine similarity between text tokens and image tokens across different layers. Figure 6 demonstrates that the modular structure of QSLAW enhances alignment capability, potentially surpassing the projector under full-precision training. However, such a advantage would be compromised without our multimodal warmup strategy. This findings highlight the importance of QSLAW's quantization-aware scale learning and multimodal warmup in achieving effective alignment between visual and language modalities. Such an improved alignment contributes to the model's overall performance and adaptability, making it a valuable approach for multimodal learning tasks.

## 5 CONCLUSION

In this paper, we are the first to investigate the potential of parameter quantization for MLLMs to reduce training overhead during VL instruction tuning. We propose a Quantization-aware Scale Learning method based on multimodal Warmup (QSLAW). QSLAW employs quantization and a minimal set of trainable scaling factors to achieve efficient VL instruction tuning. A novel modality-aware warmup is introduced to ensure that scale learning receives adequate multimodal instructional supervision while preserving its original linguistic knowledge. We validate QSLAW's effectiveness under various settings, demonstrating its excellent multimodal reasoning capabilities. QSLAW surpasses full-precision fine-tuning on ScienceQA and, for ChatBot tasks, effectively learns visual knowledge and instruction-following capabilities. Our work offers new insights into MLLM quantization and efficient VL instruction tuning, paving the way for further research into exploring the benefits of quantization and constructing more affordable VL instruction tuning methods. We hope this study will inspire additional advancements in the field of multimodal learning and instruction tuning.

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
