# OpenReview forum: "Advancing Multimodal Large Language Models with Quantization-Aware Scale Learning for Efficient Adaptation"
_acmmm.org/ACMMM/2024/Conference — MM2024 Poster_

### Official Review · Reviewer_oSGJ · 2024-05-29

**Rating:** 4
**Confidence:** 2

**Summary:**

This paper presents the first study to explore the potential of parameter quantization for multimodal large language models to alleviate the significant resource constraint encountered during visionlanguage instruction tuning.  Authors introduce a Quantization-aware Scale LeArning method based on multimodal Warmup, termed QSLAW. In the proposed method, authors prevent the quantized model from overfitting to multimodal data while ensuring stable adaptation of large models. The results seems great for downstream tasks.

**Strengths:**

See strengths.

**Limitations:**

1. Authors use GPT-4 evaluation for comparing QSLAW and QLoRA, which is not stable I think. Is it possible to use other metrics for that?
2. The research seems interesting, I think authors should provide links to Github codes.

**Suitability:**

2

---

### Official Review · Reviewer_d9kM · 2024-05-30

**Rating:** 5
**Confidence:** 3

**Summary:**

This paper addresses the challenge of parameter quantization for multi-modal large language models, aiming to reduce the extensive training demands encountered during vision-language instruction tuning while preserving original performance. It proposes a quantization-aware scale learning method that utilizes quantization and a minimal set of trainable scaling factors to achieve efficient vision-language instruction tuning. The experimental results demonstrate that models quantized using the proposed method perform on par with, or even surpass, their full-precision counterparts.

**Strengths:**

1) The paper's motivation is compelling. It identifies the significant redundancy in computation and memory burden in the current vision-language instruction tuning for multi-modal large language models (MLLLMs) and aims to address this issue.
2) It employs quantization-aware scale learning to fine-tune the quantized MLLLM, which effectively reduces quantization errors within each group.
3) It introduces a novel modality-aware warmup method that ensures MLLLMs receive precise multimodal instructional supervision and avoids potential overfitting of the LLM backbone on multimodal data in the later stages of training.
4) The authors conducted extensive experiments, including comparisons with two advanced MLLLMs, comparisons among various VL instruction tuning paradigms with examples under different multimodal instruction-following tasks, and using the powerful GPT-4 to evaluate the response quality from the proposed method and the comparative methods.

**Limitations:**

1) The authors need to pay attention to some writing details. For example, the title of Section 3.3, "QSLAW," is too abstract and difficult to understand. Additionally, in Section 4.2, the term "our QLoRA" appears, but to my knowledge, QLoRA is not a method proposed in this paper. The varying sizes of figures throughout the paper also affect the reading experience.
2) The authors did not clearly specify the definitive, general, and widely accepted evaluation metrics for assessing the proposed method. In Section 4.1 "Experimental setting," the authors mention, "We report the average accuracy on its test split," and also evaluate the responses using ChatGPT. It is uncertain whether these evaluation methods are reasonable and fair.
3) The authors mention in the title that the proposed method is efficient, but I did not find clear evidence related to efficiency in the methodology and experimental sections. Specifically, the experiments should validate why their method is considered efficient. The same requirement applies here as above.

**Suitability:**

3

---

### Official Review · Reviewer_3nEB · 2024-06-03

**Rating:** 3
**Confidence:** 2

**Summary:**

This paper ntroduces QSLAW, a Quantization-aware Scale Learning method based on multimodal Warmup, designed to enhance the efficiency of adapting multimodal large language models (MLLMs) to vision-language tasks. QSLAW addresses the challenge of significant resource constraints during MLLM tuning by introducing group-wise scale factors for quantized LLM weights, which mitigates quantization errors caused by activation outliers.

**Strengths:**

The QSLAW method can effectively save memory and preserve tuning performance.
The authors present many visual results.
The paper was well-written and organized.

**Limitations:**

I acknowledge the numerous demos provided by the author, but the article lacks a systematic description of the method part. It contains more experiential descriptions and explanations rather than detailed, structured expressions and quantitative experiments.

**Suitability:**

3

---

### Meta-Review · Area_Chair_3v1R · 2024-07-03

**Recommendation:** Accept (Poster)
**Confidence:** 3

**Metareview:**

This paper addresses an interesting topic. Following the rebuttal, all reviewers have given positive feedback, leading the AC to recommend acceptance. However, several reviewers have noted issues regarding readability and clarity of descriptions. Please revise your paper to address the reviewers' comments.